# Diagnostic Approaches for Neuroendocrine Neoplasms of Unknown Primary (NEN-UPs) and Their Prognostic Relevance—A Retrospective, Long-Term Single-Center Experience

**DOI:** 10.3390/cancers15174316

**Published:** 2023-08-29

**Authors:** Moritz Schmidt, Clemens Hinterleitner, Stephan Singer, Ulrich M. Lauer, Lars Zender, Martina Hinterleitner

**Affiliations:** 1Department of Medical Oncology & Pneumology (Internal Medicine VIII), University Hospital Tuebingen, 72076 Tuebingen, Germany; 2ENETS Center of Excellence, University Hospital Tuebingen, Otfried-Mueller-Str. 14, 72076 Tuebingen, Germany; stephan.singer@med.uni-tuebingen.de; 3DFG Cluster of Excellence 2180 ‘Image-Guided and Functional Instructed Tumor Therapy’ (iFIT), University of Tuebingen, 72076 Tuebingen, Germany; 4Cancer Biology and Genetics, Memorial Sloan Kettering Cancer Center, New York, NY 10065, USA; 5Department of Pathology, University Hospital Tuebingen, 72076 Tuebingen, Germany; 6German Cancer Consortium (DKTK), German Cancer Research Center (DKFZ), 72076 Tuebingen, Germany

**Keywords:** neuroendocrine neoplasm, cancer of unknown primary, diagnosis, immunohistochemistry, imaging, endoscopy, prognosis, overall survival

## Abstract

**Simple Summary:**

Neuroendocrine neoplasms (NENs) of unknown primary (NEN-UPs) encompass a small group within all diagnosed NENs. As a result, diagnostic work-up and the therapy algorithm are not standardized. In this study we consecutively analyzed a cohort of 113 patients, initially diagnosed as NEN-UPs. After extensive diagnostic work-up, in 11.5% (13 patients) a primary tumor site could be identified. Our study revealed that an extensive diagnostic work-up, in particular somatostatin receptor (SSTR)-PET, is associated with a better clinical outcome. Our study proposes that a detailed diagnostic investigation and the identification of a primary tumor site can provide the rationale for a tumor-specific therapy, contributing to a prolonged survival in NEN-UPs.

**Abstract:**

Neuroendocrine neoplasms (NENs) represent a rare and heterogenous group of tumors with predominantly gastroenteropancreatic or pulmonary origin. Despite numerous diagnostic efforts, the primary tumor site remains unknown in up to 20% of the patients diagnosed with NEN. In this subgroup of NEN patients, a standard diagnostic algorithm has not yet been integrated into clinical routine. Of note, an undetermined primary tumor site in NENs is associated with an impaired clinical outcome by at least “formally” limiting treatment options exclusively approved for NENs of a certain histological origin. In this retrospective study, a patient cohort of 113 patients initially diagnosed with NEN of unknown primary (NEN-UP) was analyzed. In 13 patients (11.5%) a primary tumor site could be identified subsequently, amongst others, by performing somatostatin receptor (SSTR)-PET-based imaging, which was irrespective of the initial clinical or demographic features. Diagnostic work-up and therapeutic regimens did not differ significantly between patients with an identified or unidentified primary tumor site; only a detailed immunohistochemical assessment providing additional information on the tumor origin proved to be significantly associated with the detection of a primary tumor site. Our study revealed that a profound diagnostic work-up, particularly including SSTR-PET-based imaging, leads to additional treatment options, finally resulting in significantly improved clinical outcomes for patients with NEN-UPs.

## 1. Introduction

Neuroendocrine neoplasms (NENs) constitute a rare and heterogenous group of malignancies that arise from neuroendocrine cells in almost every part of the human body. Most frequently, NENs are diagnosed in the gastrointestinal tract, the lungs and the pancreas [1]. The clinical outcome of NENs can vary widely depending on the histological grading, proliferative and endocrine activity. Based on their proliferative activity (Ki-67 rate) and degree of differentiation, NENs can be subdivided into two main subgroups: neuroendocrine tumors (NETs) and neuroendocrine carcinomas (NECs) [2]. Low-grade NETs are frequently associated with a moderate clinical course of tumor progression, which can be, however, affected by symptoms resulting from endocrine activity of the tumor. High-grade NECs, on the other hand, show rapid disease progression with the potential for early and extensive metastasis [3,4,5].

The group cancer of unknown primary (CUP) comprises numerous malignant metastatic diseases that share the feature that the primary tumor site could not be identified at initial diagnostic work-up [6]. Worldwide, up to nine percent of all newly diagnosed malignancies have been defined as CUPs. Regarding this, CUP represents one of the most common types of cancer [7,8,9,10]. Histologically, adenocarcinomas can be identified in the majority (approximately 80%) of all newly diagnosed CUP cases; the rest are represented by squamous cell carcinomas (15%) and other histological forms (5%) [6]. Based on clinical and histological features, CUP patients can be divided into two groups, which is important for further treatment and in terms of clinical outcomes [9]. The majority of these patients (80–85%) show an unfavorable prognosis with a median overall survival (OS) of only three to six months and a clinical course characterized by treatment resistance. This highlights that CUP is one of the leading malignancies responsible for cancer-related mortality. However, 15 to 20% of patients diagnosed with CUP show a more favorable prognosis with a median OS of ten to 16 months, and they can be treated similarly to specific types of cancer of a known primary tumor site. Amongst others, this group comprises patients with singular metastases of an unknown primary tumor site, females with exclusive axillary lymph node metastases and neuroendocrine carcinomas of an unknown primary tumor site [11,12,13,14,15,16,17,18,19]. Diagnostic advances have recently led to an expansion of this group by adding CUPs with the signature of colorectal, lung or renal cancer, which offers further tumor-specific treatment strategies for these patients [20].

In this context, fewer than five percent neuroendocrine neoplasms of unknown primary (NEN-UPs) form a small subgroup of CUPs [21]. Of note, among patients with NEN, a primary tumor site cannot be identified in ten to 20 percent [22,23]. Within the whole CUP cohort, NEN-UPs seem to be associated with a more beneficial prognosis with regard to OS [21,24]. Focusing exclusively on NEN, recent studies revealed that the lack of identification of the primary tumor site appears to be an unfavorable prognostic factor. However, the substantial benefit of identifying the primary tumor site in NEN-UPs has only been insufficiently studied so far [25,26,27,28,29].

For NENs, variable treatment modalities depending on disease characteristics, especially the primary tumor site, histological grading, proliferative activity and somatostatin receptor (SSTR) expression, are available. Hence, an accurate diagnostic work-up, including extended immunohistochemical analyses, PET-based imaging and endoscopic procedures, with the aim of identifying the primary tumor is required in order to enable a therapy that is optimally adjusted to the respective malignancy [22]. So far, specific systemic treatment approaches are approved only for particular subtypes of NENs. This includes the multikinase inhibitor sunitinib or the chemotherapeutic agent streptozotocin for the treatment of NENs of pancreatic origin [22,30]. However, for NEN-UPs no explicit diagnostic or therapeutic standards have been established so far.

Here, we retrospectively analyzed the diagnostic work-up performed in 113 patients with NEN-UPs and characterized the relationship between certain diagnostic work-ups and the identification of the primary tumor site. Presumably, due the low number of “only” 13 patients in the cohort with an identifiable primary tumor site, the detection of a primary tumor site was not found to be associated with a significant survival benefit. However, thoroughly conducted diagnostics, especially immunohistochemical analyses and PET-based imaging, appear to be favorable for the prognosis of patients with NEN-UPs and can contribute to extending the therapeutic potential.

## 2. Materials and Methods

### 2.1. Study Population

Patients initially diagnosed with a neuroendocrine malignancy of unknown primary who presented between January 2010 and May 2023 at the University Hospital Tuebingen were considered for this observational study. Finally, a total of 113 patients (63 males and 50 females, with a mean age of 63.4 ± 13.5 years) were included in this retrospective chart review. Patient characteristics are shown in detail in Table 1. The study was approved by the IRB (ethics committee of the Faculty of Medicine of the Eberhard Karls University Tuebingen) of the University Hospital Tuebingen (reference number 109/2023BO2) and was conducted in accordance with the Declaration of Helsinki.

### 2.2. Collection of Data

For each patient included in the study, the following parameters were evaluated: sex, age, date of initial diagnosis, primary diagnosis, OS after initial histological diagnosis of NEN, histological subtype, TNM classification, UICC stage, origin of initial histology, histological grading, proliferation index, number and distribution of metastatic sites, body mass index (BMI) and secondary diagnoses. The absence of distant metastases was defined as (I) the sole presence of lymph node metastases or (II) the exclusive presence of a singular NEN manifestation that did not appear to correspond to the primary tumor site clinically, as well as histologically. The diagnostic work-up included the determination of the absolute neutrophil count (ANC), absolute lymphocyte count (ALC), hemoglobin (Hb) level, platelet (PLT) and white blood count (WBC), serum level of chromogranin A (CgA), neuron-specific enolase (NSE), carcinoembryonic antigen (CEA), CA 19-9, lactate dehydrogenase (LDH) and 24 h urine level of 5-hydroxyindoleacetic acid at initial diagnosis. Additionally, immunohistochemical analyses of the initially extracted tissue specimens were performed with regard to the biomarkers CDX2, TTF-1, ISLET-1, synaptophysin, CgA, CD56, serotonin, gastrin and cytokeratin 20 (CK20). Based on the immunohistochemical staining for CDX2, TTF-1, ISLET-1, synaptophysin and CgA, patients were classified into the following categories: “no adherence” (no marker stained), “minimal adherence” (one or two markers stained), “partial adherence” (three or four markers stained), “full adherence” (five markers stained). About 50% of the patient cases have been histopathologically diagnosed at the Pathology Institute of the University Hospital Tuebingen. For those patients, the manual counting of Ki67 of hotspot areas was performed, and since 2021, digital Ki67 evaluation was also performed (using Cognition Master, VMscope GmbH, Berlin, Germany). We used the Ki67 (clone Mib1) antibody from Agilent Technologies (Santa Clara, CA, USA) (#M724, 1:400). The mitotic index was not routinely assessed (except for pulmonary typical and atypical carcinoids). Furthermore, diagnostic tools, including imaging methods (18F-FDG PET and SSTR-PET imaging), endoscopy (gastroscopy and colonoscopy), next-generation sequencing (NGS) and an analysis of chromosomal instability (CIN), were considered. Summarizing the entity of the diagnostic work-up, patients were sectioned according to the performance of immunohistochemical and endoscopic analyses, PET-based imaging and the investigation of serum tumor markers (CgA and NSE). Patients were assigned into the subgroup “Primary identified (PI)” only after explicit diagnosis within a medical document. Patients who were not diagnosed with an identified primary tumor site were classified as “NEN-UP”. Based on a histological discrimination, patients were divided into the subgroup “Neuroendocrine tumor (NET)” or “Neuroendocrine carcinoma (NEC)”.

On top of that, treatment algorithms applied to each patient case were assessed, in special treatment regimens, including systemic therapy (chemotherapy, somatostatin analogues (SSA), immunotherapy and targeted therapy), radiation therapy (local radiotherapy, peptide receptor radiotherapy (PRRT) and selective internal radiation therapy (SIRT)), local therapy (surgery and radiofrequency ablation (RFA)) and best supportive care (BSC), as well as the sequence and number of therapies.

### 2.3. Data Analysis

Descriptive statistics were used for distinguishing patients according to age, sex, histological grading, proliferation index, number and distribution of metastatic sites, origin of initial histology, date of initial diagnosis and BMI. Patients classified into the subgroup PI or NEN-UP were contrasted with respect to the abovementioned properties. Patients belonging to the subgroups PI or NEN-UP and NET or NEC, respectively, were compared with regard to serum levels of CgA, NSE, CEA, CA 19-9, LDH and Hb, as well as WBC, ANC, ALC and platelet counts. Furthermore, a comparison between these subgroups was realized focusing on the abovementioned immunohistochemical markers, PET-based imaging, endoscopic analyses and the performance of NGS and analyses of CIN. In addition, therapy characteristics were compared between the subgroups PI and NEN-UP.

The statistical significance of differences was analyzed using the chi-squared or Fisher’s exact test for categorical variables and the unpaired *t*-test for continuous variables. The predictive value of CgA, NSE and LDH serum levels were evaluated by examining the area under the receiver operator characteristic (ROC) curve with a confidence interval of 95%. A univariable logistic regression analysis was performed to assess any association between diagnostic tools and identification of the primary tumor site. OS was calculated using the Kaplan–Meier method, and the log-rank test was used to compare differences between subgroups. Survival curves were created for NEN-UP vs. PI, NET vs. NEC, histological grading, diagnostic lines, endoscopy, immunohistochemical analyses, PET-based imaging, therapy lines, CgA, NSE and LDH serum levels. Continuous variables are presented as the mean, and categorical variables are given by numbers and percentages. All statistical tests were considered statistically significant when *p* was below 0.05. Statistical analysis was performed using GraphPadPrism (v.9.4.1) (Dotmatics, Boston, MA, USA).

## 3. Results

### 3.1. Demographic and Clinical Characteristics of Patients with NEN-UP

In this study, a total of 113 patients with initially diagnosed NEN-UP were retrospectively analyzed (Table 1). The mean age was 63.4 years, and 55% of the patients were male. The average body-mass-index (BMI) was 25 kg/m^2^. The predominant histological subtype was NEC (38%), followed by NET G2 (31%), NET G3 (15%) and NET G1 (14%), with a mean proliferation rate of 36%. The most frequent sites of distant metastases were the liver (68%), lymph node (54%) and bone (29%), and the majority of patients initially showed either one (33%) or two metastatic sites (30%). The diagnosis of more than half of patients (57%) was based on a tissue sample derived from the liver. In 13 out of 113 patients (11%) of our study cohort, a primary tumor could be identified subsequently. Apart from the origin of the initial histology (*p* = 0.047), no statistically significant association could be found between the aforementioned characteristics of the patients and the identification of a primary tumor site (Table 1). Within the cohort of patients diagnosed with NET or NEC, eight (11%) or five (11%) patients had a primary identified tumor (Figure 1A). The primary could be detected in most of the cases in the midgut (38%), followed by the lungs (30%) and the pancreas (15%) (Figure 1B). The respective fraction of the histological subtypes within the PI and NEN-UP cohort appeared rather similar and is shown in Figure 1C. Additionally, lymph node invasion, the presence of distant metastases and the UICC stage were assessed and are shown in Figure 1D. In summary, patients with an identified primary tumor site seemed to suffer from NENs in more advanced stages as revealed by the occurrence of distant metastases and the UICC stage compared to the NEN-UP cohort.

### 3.2. Assessment and Comparison of the Diagnostic Work-Up Performed in Patients with NEN-UP

The diagnostic work-up was assessed with regard to hematological parameters, immunohistochemical analyses, PET-based imaging, endoscopy, NGS and an analysis of CIN (Table 2 and Table 3). Blood parameters included the determination of standard parameters (blood count and LDH) and serum tumor markers. Altogether, blood counts and LDH were determined in 77% of the patients at the initial diagnosis. For serum tumor markers, ratios of completed diagnostics varied among the patients, with CgA being the most frequently (66%) and CEA (24%) the least frequently assessed marker.

In addition, 24 h 5-hydroxyindoleacetic acid excretion was only assessed in a total of 5% of the patients that were exclusively belonging to the NEN-UP cohort.

Regarding immunohistochemical analyses, performance rates were variable between the markers listed below. The markers synaptophysin, CgA and CD56, indicating neuroendocrine differentiation in general, were stained in 81%, 65% or 36%, respectively. Among CDX2, TTF-1 and ISLET-1, which correlate with an intestinal, pulmonary or pancreatic origin of the tumor, the expression of TTF-1 was most frequently determined (67%), followed by CDX2 (58%) and ISLET-1 (35%). The endocrine markers serotonin and gastrin were stained only in a minority of patients (15% and 2%). Staining of CK20, possibly indicating the presence of Merkel cell carcinoma, was performed in 25% of the patients.

The staining of the immunohistochemical markers CDX2, TTF-1, ISLET-1, synaptophysin and CgA in the patients was distributed as follows: in 28% of all patients, all markers were stained. Three or four markers and one or two markers were analyzed in 35% or 23% of the patients, respectively. None of these markers were analyzed in 12% of the patients indicating that these patients were initially diagnosed solely by means of morphological assessments in histopathology in combination with clinical, laboratory and imaging findings.

Moreover, 82 out of 113 patients (72%) received any kind of PET-based imaging: SSTR-PET was performed in 71 patients (62%) and FDG-PET was applied to 24 patients (21%).

Endoscopy-based diagnostic approaches (gastroscopy and/or colonoscopy) were performed in 60 out of 113 patients (53%) at the initial diagnosis. Herein, 70% of patients received both gastroscopy and colonoscopy, and 11 and 7 patients received gastroscopy or colonoscopy alone.

Additional diagnostics using NGS were performed in 31 patients (27%), and genetic alterations detected using these means are shown in Appendix A. An analysis of CIN was performed in 30 patients (26.5%).

The degree of diagnostic efforts performed at the primary diagnosis was rated based on the performance rates of (I) immunohistochemical and (II) endoscopic analyses, (III) PET-based imaging and the investigation of (IV) serum tumor markers (CgA and NSE). All four diagnostic tools were used in “only” 35% of the patients; 30%, 22% and 9% of the patients were diagnosed based on three, two or one of the aforementioned diagnostic instruments, respectively. In 3 out of 113 patients (2%), none of these diagnostic methods were recorded according to our files.

In the next step, the diagnostic work-up was compared between the cohorts NEN-UP and PI, on the one hand, and the cohorts NET and NEC, on the other hand (Table 2 and Table 3). In a comparison of the PI and the NEN-UP cohort, statistically significant differences could be detected regarding the performance rates of serum levels of NSE (*p* = 0.003), LDH (*p* = 0.037), neutrophils and lymphocytes (*p* = 0.031 each) and the immunohistochemical analysis of CgA (*p* = 0.033) and CK20 (*p* = 0.020). Besides the immunohistochemical staining of CK20, these characteristics were more deeply investigated within the cohort PI.

Between the NET and NEC cohort, several differences in the performance of diagnostics could be observed. The determination of serum levels of CgA, as well as the immunohistochemical analysis of ISLET-1 and serotonin and performance of SSTR-PET, was statistically more often performed in patients belonging to the NET cohort. Patients diagnosed with NEC, in contrast, were subjected to an immunohistochemical analysis of CD56 and FDG-PET-based imaging more commonly. On top of that, diagnostic work-up was more thoroughly realized in patients with NETs.

Comparing the absolute values of the blood parameters CgA, NSE, LDH, Hb, WBC, ANC, ALC and PLT between patients with a NEN of a known or unknown primary, no statistically significant differences could be found between the two subgroups (Figure 2).

Within the subgroups NET and NEC, statistically significant differences could be observed between the determination of serum levels of CgA (*p* = 0.013), the immunohistochemical assessment of ISLET-1 (*p* = 0.043), CD56 (*p* = 0.005) and serotonin (*p* = 0.015) and FDG- (*p* = 0.001) or SSTR-PET-based imaging (*p* < 0.0001). CgA, ISLET-1 and serotonin were significantly less frequently assessed in the sera or tumor tissues of patients belonging to the NEC cohort, whereas CD56 was significantly more often stained in tumor tissues of patients with NEC. Furthermore, SSTR-PET performance rates were significantly higher and FDG-PET performance rates were significantly lower in the NET cohort. Finally, a statistically significant difference concerning the extent of diagnostics performed could be found between the two subgroups suggesting a more detailed work-up within the NET cohort.

### 3.3. Association and Predictive Value of Diagnostic Instruments and Detection of Primary Tumor Sites in Patients with NEN-UP

We further evaluated the association between the diagnostic instruments and the detection of a primary tumor site by means of univariable logistic regression analysis (Table 4). We identified that the assessment of serum levels of NSE (OR 13 [2.4–241.2]; *p =* 0.016) and the adherence to immunohistochemical guidelines (OR 1.91 [1.21–3.48]; *p =* 0.015) were statistically significantly associated with the detection of a primary tumor site within our patient cohort.

Additionally, we analyzed the predictive values of CgA, NSE and LDH for the detection of a primary tumor site by means of ROC analysis (Figure 2B). In summary, the predictive value of these parameters showed only limited accuracy (AUC 0.56–0.68).

### 3.4. Treatment Strategies for Patients with NEN-UP

Treatment characteristics are illustrated in Table 5. Here, 104 patients (92%) were treated with either systemic cancer therapy, radiation therapy or local therapy, and 8 patients (7%) obtained BSC at the initial diagnosis. Further, 91 patients (80%) underwent systemic cancer treatment, including conventional chemotherapy (58 patients, 51%), therapy with SSA (42 patients, 37%), targeted therapy (8 patients, 7%, represented by the multikinase inhibitor sunitinib and the mTOR inhibitor everolimus) and immunotherapy (4 patients, 3%), and 26 patients (23%) underwent local radiotherapy; 25 patients (22%) received PRRT, and SIRT was applied to 4 patients (3%). Surgical procedures were realized in 39 patients (34%), and RFA was performed on one patient (1%). The majority of patients underwent one line of treatment, and herein, platinum-based chemotherapy and SSA therapy represented the most common treatment regimens. Further details on treatment modalities in this patient cohort are depicted in Appendix A.

Statistically significant differences between the cohorts NEN-UP and PI were only found in the context of PRRT (*p =* 0.008) and the number of systemic therapy lines (*p =* 0.013). A higher portion of patients with an identified primary tumor site appeared to be treated with PRRT. Furthermore, these patients were found to receive a second-line systemic therapy more frequently than patients with an unidentified primary tumor site.

### 3.5. Clinical Outcomes of Patients with NEN-UP

Finally, we investigated the prognostic impact of clinical characteristics, diagnostic procedures and therapeutic approaches on the clinical outcomes of patients with NEN-UP. The median OS for the entire cohort was 57 months (Figure 3A). Of note, we observed a long-term OS survival benefit in the PI group. However, due to the small sample size in our cohort, this OS benefit failed to be statistically significant (median OS of 101 months in the PI cohort vs. median OS of 37 months in the NEN-UP cohort, *p* = 0.364, Figure 3B). This observation could be confirmed when investigating the OS in patients with and without an identified primary tumor site within the NET subgroup (Appendix A; median OS of 101 months in the PI cohort vs. median OS of 70 months in the NEN-UP cohort within the NET subgroup, *p* = 0.158).

As expected, histological grading was shown to be a prognostic marker in patients with NEN-UP: patients suffering from NET G2 had a more favorable prognosis, compared to patients diagnosed with NEC who had the least favorable prognosis with regard to OS (median OS of 74 months in the NET G2 cohort; median OS of 10 months in the NEC cohort, *p* < 0.0001) (Figure 3C).

Of note, the extent of the diagnostic work-up, defined based on performance rates of serum tumor markers, immunohistochemistry, endoscopy and PET-based imaging, was clearly associated with clinical outcomes of patient prognosis (25 months for the median OS in the cohort receiving two diagnostic lines vs. 10.5 months for the median OS in the cohort receiving no diagnostic line, *p* = 0.029) (Figure 3D). Among the diagnostic work-ups, only PET-based imaging seemed to have a significant impact on the prognosis of patients with NEN-UP, whereas immunohistochemical analyses and endoscopy showed only a minor impact on the OS of the patient cohort (Figure 3E–G). Patients receiving PET-based imaging, in general, showed a significantly prolonged OS compared to patients without any kind of PET-based imaging (median OS of 74 months in the PET cohort; median OS of 10 months of the cohort without access to PET, *p* < 0.0001) (Figure 3G).

The performance of endoscopy or an extensive immunohistochemical analysis did not significantly correlate with prolonged survival for patients with NEN-UP (Figure 3E,F; *p* = 0.092 for endoscopy). Of note, within the PI cohort, additonal PET-based imaging was the major diagnostic tool for the identification of a primary tumor site, irrespective of a previously performed immunohistochemical analysis (eight patients, 61%). Thus, resection of the primary tumor site, thereby preventing local tumor-related complications, as well as PRRT could be realized in seven out of eight cases (87%) (Figure 4). Additionally, we assessed the impact of either FDG- or SSTR-PET-based imaging on the OS in the entire NEN cohort. SSTR-PET, but not of FDG-PET, was significantly associated with a prolonged OS (median OS of 74 months in the SSTR-PET cohort; median OS of 10 months in the cohort without access to PET, *p* < 0.0001) (Appendix A).

Next, blood-based parameters were categorized and correlated with the OS of patients with NEN-UP: for CgA and NSE, a serum level exceeding double the regular serum concentration (102 µg/L for CgA and 12.5 µg/L for NSE), and for LDH, a serum concentration higher than 250 U/L was defined as “high”, lower values were defined as “low”, respectively. According to this subdivision, high serum levels of LDH and NSE, but not of CgA, were significantly associated with an unfavorable prognosis regarding the OS (median OS of 7 months in the LDH-high cohort vs. median OS of 101 months in the LDH-low cohort, *p* < 0.00001; median OS of 57 months in the CgA-high cohort vs. median OS of 101 months in the CgA-low cohort, *p* = 0.0521; median OS of 9 months in the NSE-high cohort vs. median OS of 101 months in the NSE-low cohort, *p* = 0.0001) (Figure 3H–J). A similar trend could be seen within the subgroups NEN-UP and PI in the context of LDH and NSE levels, although higher NSE levels were not statistically significantly associated with an impaired OS within the PI cohort. Regarding CgA, again no statistically significant association between OS and distinct serum levels could be stated (Appendix A).

Finally, the relationship between the number of therapy lines and the OS within the entire patient cohort is depicted in Appendix A.

## 4. Discussion

In this retrospective study, we consecutively analyzed a heterogenous group of patients with the initial diagnosis of neuroendocrine neoplasms of unknown primary (NEN-UP), regarding the clinical characteristics, diagnostic work-up and therapeutic treatment. Our aim was (I) to identify a common diagnostic work-up that can lead to the detection of a primary tumor site and (II) to assess which of these diagnostic procedures could contribute to improve the clinical outcome of these patients.

Between 2010 and 2023, 1325 patients diagnosed with NEN were treated at the University Hospital Tuebingen, and among these patients, a primary tumor site could not be detected in 113 patients (8%) at the initial diagnosis. This is in accordance with the previously described prevalence of CUP in patients with NENs [22,23]. Subsequently, a primary tumor site could be identified in 13 out of these 113 patients (11%). The detection rate of a primary tumor site in patients originally being classified as having a CUP is known to vary between 10 and 40% according to the literature and could be increased more recently by means of newly available diagnostic tools, including NGS or molecular profiling [31,32,33,34,35]. Since targeted therapy approaches are available, an accurate characterization of the tumor is essential for the detection of potentially targetable molecular alterations. At this point, recent studies revealed that diagnostic tools focusing on epigenetic biomarkers, and especially DNA methylation signatures, have the potential to identify the primary tissue of the tumor in patients diagnosed with CUP with a higher likelihood compared to other diagnostic instruments [36,37]. The assignment of the tumor to a specific tissue by means of epigenetic profiling points to certain genetic alterations typical for this type of cancer and relevant to therapy, which can be specifically investigated in further diagnostic studies [38]. Thus, therapeutic options can be increased, which can contribute to improving the clinical outcomes of patients with CUP. However, these studies were mostly performed on patients with different histological types of CUP, and herein, patients with NEN-UP represented only a minority.

The majority of patients (61%) included in our study histologically showed a NET of different grading, whereas 39% belonged to the group of NECs. Previous publications describe the diverging histological backgrounds of patients with NEN-UP, underlining the biological heterogeneity of this patient cohort [25,28,29]. Nonetheless, the portion of patients with an identified primary appeared to be quite similar within the two patient cohorts (11.4% in the NET cohort vs. 11.6% in the NEC cohort). We found out that the midgut was the most common site where a primary could be detected, which is in line with previous data [23,29,39]. However, within different primary tumor sites, the midgut comprises a relative fraction of 38%, which is considerably lower compared to other studies. This could be due to the small cohort size of patients with an identified primary tumor site or the relatively high portion of patients with a histologically diagnosed NEC, which mainly did not arise from the small intestine.

For NEN-UPs however, no standard diagnostic procedure is established so far. According to the ENETS Consensus Guidelines Update and the German practice guideline for neuroendocrine tumors, additional diagnostic tools, such as endoscopy, PET-based imaging, expanded immunohistochemical analysis and the determination of urine levels of 5-hydroxyindoleacetic acid at initial diagnosis, should be included in the work-up to identify the primary tumor site [22,40]. These diagnostic tools were variably implemented in our patient cohort—SSTR-PET-based imaging, for example, was performed in 71 patients, whereas 46% of patients did not receive an endoscopic assessment. Interestingly, we identified only several significant differences between patients with or without an identified primary tumor site with regard to the performance rates of certain diagnostic approaches. This is in line with a study performed by Wong and colleagues, investigating the role of diagnostic work-up in the identification of a primary tumor in a patient cohort with CUP of different histological backgrounds [41]. Referring to this, the authors could not find any statistically significant differences between patients with an identified or an unidentified primary. However, patients with tumors with neuroendocrine differentiation comprised only 15% of the whole patient cohort.

In NEN-UPs, the investigation of additional immunohistochemical markers appears feasible and can enable a more precise attribution to a primary organ [2]. This is also reflected by our finding that no specific marker, but a thoroughly performed immunohistochemical work-up, significantly increases the probability of identifying the primary tumor site. In 12% of all patients with NEN-UPs, no immunohistochemistry at initial diagnosis was documented in our system; 85% of these patients did not receive their histological diagnosis at the University Hospital Tuebingen and subsequently presented for further treatment planning at our center. Even despite our great effort to obtain all histopathological data, no dedicated IHC staining, including CgA, synaptophysin and CD56, even if performed elsewhere, could be verified. Only one of these patients was initially diagnosed at the University Hospital Tuebingen. In this case, unfortunately, the biopsy was not suitable for extensive IHC analysis. As a result, the initial diagnosis was based on a morphological assessment of histopathology in combination with clinical, laboratory and imaging findings. In patients who did not receive a dedicated IHC analysis to further identify the primary organ, three possible reasons were identified: (I) not enough tumor material; (II) lack of clinical evidence for a NEN-UP; (III) conservation of the material for molecular pathology. Nonetheless, these markers show limited sensitivity and specificity in general, and in particular, in poorly differentiated NEC, they can be expressed divergently, which has to be considered in clinical decision-making [28,42,43]. Based on this and the fact that 43 out of 113 patients are diagnosed with NEC in our cohort, it is possible that individual markers are not sufficient for the identification of a primary tumor site here and performance rates of the respective markers have to be considered in the context of the histological subtypes. Here, ISLET-1 was used as an immunohistochemical marker suggestive of a pancreatic origin of NENs. However, ISLET-1 has limited specificity for pancreatic NENs. On the one hand, poorly differentiated NENs of pancreatic origin show lower expression of ISLET-1. On the other hand, the expression of ISLET-1 was also detected in extrapancreatic NENs, like for example in neuroendocrine carcinomas of the lungs, of the head and neck or in Merkel cell carcinomas [43]. Based on this, ISLET-1 was more recently seen as a general marker of neuroendocrine differentiation, which, compared to the conventional markers CgA and synaptophysin, is also stably expressed in many high-grade NECs [44]. These aspects must be considered when interpreting our data regarding the potential tissue of origin of NEN-UPs. In the meantime, additional immunohistochemical markers with a higher specificity for pancreatic NENs could be identified, like PDX1, PAX8 or NESP55 [45]. In this context, however, no other marker besides ISLET-1 was established and available at the University Hospital of Tuebingen and pathology institutes of external referring physicians during the observation period from 2010 to 2023.

Appropriate imaging is crucial in NEN, especially without an identified primary tumor, for determining the extent of the malignancy at the primary diagnosis, therapy evaluation and control, as well as follow-up. Besides conventional full-body CT scans, which have been completed in all of the patients in this retrospective study, hybrid imaging techniques are used especially for staging purposes. SSTR-PET is mainly applied to low-grade NENs and can provide information with regard to a potential SSTR-directed therapy, whereas FDG-PET-based imaging is more suitable to high-grade NENs [46]. Dividing our patient cohort into low-grade (NET G1 and NET G2) and high-grade (NET G3 and NEC) neoplasms, SSTR-PET was performed in 49 out of 52 patients (94%) in the low-grade cohort, and FDG-PET was applied to 19 out of 61 patients (31%) indicating a discrepancy in terms of hybrid imaging to the disadvantage of high-grade NENs. Of note, SSTR-PET was even more frequently performed in the high-grade patient cohort compared to FDG/PET-CT (22 out of 61 patients, 36%). In this subgroup, we observed that the same number of patients in the NET G3 and NEC cohort underwent SSTR-PET (11 patients, respectively). In contrast, 16 patients with NEC, but only three patients with NET G3, received FDG-PET. At first glance, the performance of SSTR-PET in NEC appears counterintuitive. Nonetheless, previous publications report on an expression rate of SSTRs between 32 and 50% in NECs. For NET G3, the expression of SSTR could even be detected in 67–96% [47,48,49,50]. These insights make it understandable why patients with high-grade NENs are also subjected to SSTR-PET. Based on the expression of SSTR, in principle, these tumors can be targeted by means of PRRT. Recent studies could already confirm the therapeutic benefit of PRRT in SSTR-positive NET G3, in particular in patients showing a proliferative index of maximally 55% [51,52,53]. According to the literature, the sensitivity of SSTR-PET for the diagnosis of NEN accounts for 36–60% in patients with NEN-UP, and for FDG-PET, it varies between 37 and 72% in NEN in general [54,55,56,57,58,59]. On the one hand, our data revealed that PET-based imaging was not significantly associated with identifying a primary in our cohort according to the performed univariable logistic regression—on the other hand, within the PI cohort, we could identify PET-based imaging as a major tool for the identification of a primary tumor (Figure 4).

Endoscopy plays an important role in identifying the primary tumor site, especially if the primary tumor is located in the upper or lower gastrointestinal tract. The sensitivity of upper and lower gastrointestinal endoscopy varies between 50 and 80%, which cannot be generalized for all types of NEN, since these procedures are mostly investigated in patients belonging to distinct histological subgroups, for example in patients with gastric or pancreatic NEN [60,61]. For NEN-UPs, it is suggested that most of the occult primaries are localized in the small intestine [23]. For the detection of primary tumors of the small intestine, costly techniques, such as video capsule endoscopy or double-balloon enteroscopy, are required, which are not widely available to date. Previously published reports highlight the potential of these diagnostic tools in identifying the primary tumor site in NEN-UPs. In this regard, sensitivity varied between 33 and 88% and 60 to 80% for double-balloon enteroscopy and capsule endoscopy, respectively [51,62,63,64,65]. In our patient cohort, at least in one out of 13 patients (7%), a primary could be identified by means of endoscopy, although univariable logistic regression analysis could not detect any significant association between the performance of different endoscopic approaches and the detection of a primary, and endoscopy was performed quite inconsistently in this patient cohort.

Additionally, biochemical markers found in the blood or urine of patients may contribute to the diagnosis and follow-up of NENs. Here, we focused on the serum tumor markers CgA and NSE, as well as 5-hydroxyindoleacetic acid, as a central metabolite of serotonin. With a sensitivity and specificity ranging between 60 and 90%, CgA represents a crucial blood parameter in NENs of different origins, irrespective of the endocrine activity of the tumor. NSE is of particular importance in high-grade NETs and NECs, and 5-hydroxyindoleacetic acid has diagnostic value in the context of metastasized NENs of the small intestine [66]. Due to their unspecific release by different histological types of NEN, CgA and NSE can contribute to the identification of a primary tumor only to a limited extent, whereas urine analysis of 5-hydroxyindoleacetic acid shows a specificity of 89–100% depending on the cut-off value for NENs in the small intestine [67,68]. This is in line with our results obtained from ROC analyses showing the limited accuracy of CgA and NSE regarding the detection of a primary tumor site. With an analysis in only 6 out of 113 patients (5%), the determination of the 24 h urine level of 5-hydroxyindoleacetic acid has underrepresented diagnostic potential in this cohort. Besides distinct endocrine diagnostic tests for functionally active NENs, further biomarkers with higher expected specificity, like pro-gastrin-releasing peptide, pancreastatin or neurokinin A or platelet-expressed synaptophysin, are currently under preclinical investigation [69,70,71,72,73,74]. Based on this perspective, the investigation of circulating tumor cells and gene transcripts may further amplify the diagnostic repertoire, also in the context of NEN [75,76].

Similar to diagnostic approaches, no standard algorithms define the treatment of NEN-UPs so far. In most of the cases, NEN-UPs are treated along the lines of metastatic NENs with a known origin and the respective degree of differentiation. In NENs, the histologic origin has a strong impact on the therapy strategy, since certain treatment modalities are only available for distinct subentities leading to a superior clinical outcome in these patients [21]. Therefore, the identification of a primary appears pivotal for an optimal therapeutic approach. However, attention has to be paid when interpreting the results of available studies contrasting site-specific therapy with empiric chemotherapy in patients with CUP. In general, these studies are scarce, and only a minority has a prospective and randomized design. Furthermore, problems arose in recruiting patients, which delayed the completion of the studies. This highlights the deficiencies of existing studies in this field and clearly shows the need for alternative trials. Rassy et al. proposed two prospective trial designs that aim to circumvent these difficulties in order to improve the treatment and clinical outcomes of patients with CUP [77].

Within this patient cohort, 80% of the patients obtained systemic cancer treatment, which could be further subdivided into two major treatment cohorts: low-grade NEN-UPs were treated with SSA, and high-grade NEN-UPs underwent platinum-based chemotherapy. In the context of SSA therapy, both octreotide long-acting repeatable and lanreotide autogel are approved for the treatment of NEN-UPs. In our cohort and among the patients treated with SSA, 76% received lanreotide autogel and 24% underwent therapy with octreotide long-acting repeatable. Local radiotherapy and surgery were performed to reduce the symptom burden or in order to treat or prevent tumor-associated complications. Previous studies point to a benefit resulting from the surgical resection of singular abdominal metastases, especially liver metastases, in patients with NEN-UP. Of note, beside the reduction in the initial tumor burden, the primary NEN with a gastroenteropancreatic origin could be identified in up to 89%, intraoperatively [60,78,79]. Depending on the expression status of SSTR, which can be assessed by means of a pre-therapeutic SSTR-PET, PRRT was shown to be a lucrative treatment option, with response rates of approximately 70% in patients with NEN-UP [80]. Here, nearly one fourth of the patients received PRRT.

The median OS in this patient cohort was 57 months, which is superior to previously published survival data of patients with NEN-UPs, ranging from 11.6 to 40 months [24]. This could be related to a lower portion of low-grade NENs in these studies compared to our patient cohort and, since this review was published in 2011, to improvements in the management of NENs over the last years. Of note, after the identification of a primary tumor site, patients showed a more favorable clinical outcome defined based on OS (median OS of 101 months in the PI cohort vs. median OS of 37 months in the NEN-UP cohort). However, these findings were not statistically significant, probably due to the small cohort size of the PI cohort. This observation is in contrast to a previous study performed by Wong et al., who could not identify a survival benefit for patients after the identification of a primary tumor site [41]. Nonetheless, this study mainly focused on CUP patients with adenocarcinoma or squamous cell carcinoma, and NEN patients represented only a minority within the whole study cohort.

In this study, we could not find any statistically significant differences in the extent of the diagnostic work-up, defined based on the determination of serum tumor markers, immunohistochemistry, endoscopy and PET-based imaging, performed in NEN patients with or without an identified primary tumor. This could be related to the small number of patients within the PI cohort. We assume that the more favorable clinical outcome of the PI cohort goes along with the survival benefit, which is associated with a more extensive diagnostic work-up. Herein, patients receiving SSTR-PET were found to have a significantly advantageous clinical outcome. One explanation for these observations might be that an extensive diagnostic work-up, including SSTR-PET, increases the detection rate of the primary tumor site and a consecutive surgical intervention. Resection of the primary tumor is known to be associated with a prolonged median survival, at least in patients suffering from NET G1 and G2 with symptomatic liver metastases [81]. In addition, there is some evidence that resection of a NEN of pancreatic origin may also be beneficial for the survival of patients who have concomitant unresectable liver metastases [82,83]. The authors hypothesized that resection of the primary tumor alone would curb tumor growth and cancer progression. In particular in the case of NENs of the midgut, the resection of the primary tumor finally might prevent tumor-associated local complications, like ischemia or stenosis. On top of that, previous studies have also shown that surgical resection of the primary tumor leads to a better response to PRRT for different entities of NEN [84,85]. The lower tumor burden after tumor resection is assumed to result in a higher concentration of each tumor manifestation and thus a better efficacy of PRRT. Here, with the use of SSTR-PET-based imaging, the primary tumor could be identified and resected in seven patients in this cohort. Subsequently, four of these seven patients underwent PRRT after resection of the primary tumor. Furthermore, diagnostics can help to better characterize the tumor. This information can be used for therapeutic purposes by identifying patients that can undergo targeted treatment approaches, like SSA therapy and PRRT.

Nonetheless, our study has some limitations: the retrospective characteristic of the study only makes it possible to draw associations and not causal relationships between individual factors. Our initial goal was to illustrate the states regarding the diagnostic approach for NEN-UPs at our center during the last decade. Based on this, we are now planning prospective studies with the aim of evaluating the results obtained in this work. Additionally, patients were assigned to the PI cohort if a tumor manifestation was defined as a primary tumor site in a medical document. However, this is open to interpretation and may have had an impact on the distribution of patients with respect to the two cohorts. Furthermore, the number of patients examined in this study was relatively small. Despite an observation period of 13 years, a total of only 113 patients could be included in this study. This is related to the fact that NENs are currently still considered to be rare tumor diseases in Germany, although a considerable increase in the incidence of NENs could be observed recently. More precisely, Das et al. reported an incidence rate for NENs of gastroenteropancreatic origin of 2.5 cases/100,000 until 2006 in Germany [86]. Among these patients, NEN-UPs form a small subgroup, which may explain the small number of patients included in this study. Therefore, multicenter studies are needed to gain deeper insights into this research area.

## 5. Conclusions

Taken together, our work critically evaluated diagnostic procedures and their relevance for the clinical outcomes of patients with NEN-UP. Only an extensive immunohistochemical investigation, including the markers CDX2, TTF-1 and ISLET-1, was shown to be significantly associated with the identification of a primary tumor site. Due to a limited number of patients, this study could not clearly confirm that patients with NEN-UP and a subsequently identified primary tumor site show significantly improved overall survival. However, we provide evidence for a better clinical outcome of these patients compared to patients with an unidentified primary tumor site. For these patients, a complete diagnostic work-up, including serum tumor markers, immunohistochemistry, endoscopy and PET-based imaging, may contribute to improve clinical outcomes. Based on our findings, patients with NEN-UP might undergo SSTR-PET in order examine the opportunity of PRRT. Furthermore, SSTR-PET can contribute to the identification and the consecutive surgical resection of the primary tumor site. Thus, local tumor-related complications can be prevented. Nevertheless, our study has some limitations, including the retrospective and single-center study design and the relatively small number of patients, especially in the PI cohort. Therefore, our findings need to be further evaluated by means of prospective, multicenter investigations in the future. Our single-center study could provide an orientation for clinicians regarding the diagnostic work-up that should be fulfilled in patients with NEN-UP in order to identify the primary tumor site. Definitely, large-scale randomized studies are strongly required for establishing an evidence-based standard of diagnostics for patients diagnosed with NEN-UP.

## Figures and Tables

**Figure 1 cancers-15-04316-f001:**
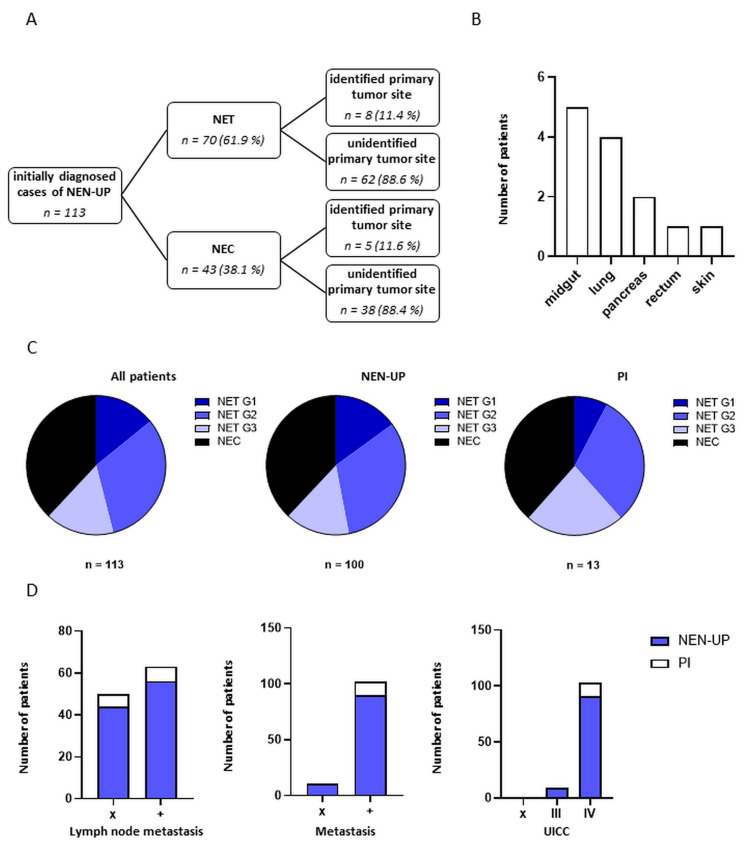
Clinical characteristics of the patient cohort. (**A**) Distribution of patients with unidentified and identified primary tumor sites within the NET and NEC cohort. (**B**) Tissue-related identification of primary tumor sites. (**C**) Distribution of histological subtypes within the entire patient cohort and patients with or without identified primary tumor sites. (**D**) Determination of lymph node invasion, distant metastases and the UICC stage. Abbreviations: NEN-UP = neuroendocrine neoplasm of unknown primary; NET = neuroendocrine tumor; NEC = neuroendocrine carcinoma; PI = primary tumor site identified later on; x = not assessable; + = present; III: presence of lymph node metastases; IV: presence of distant metastases.

**Figure 2 cancers-15-04316-f002:**
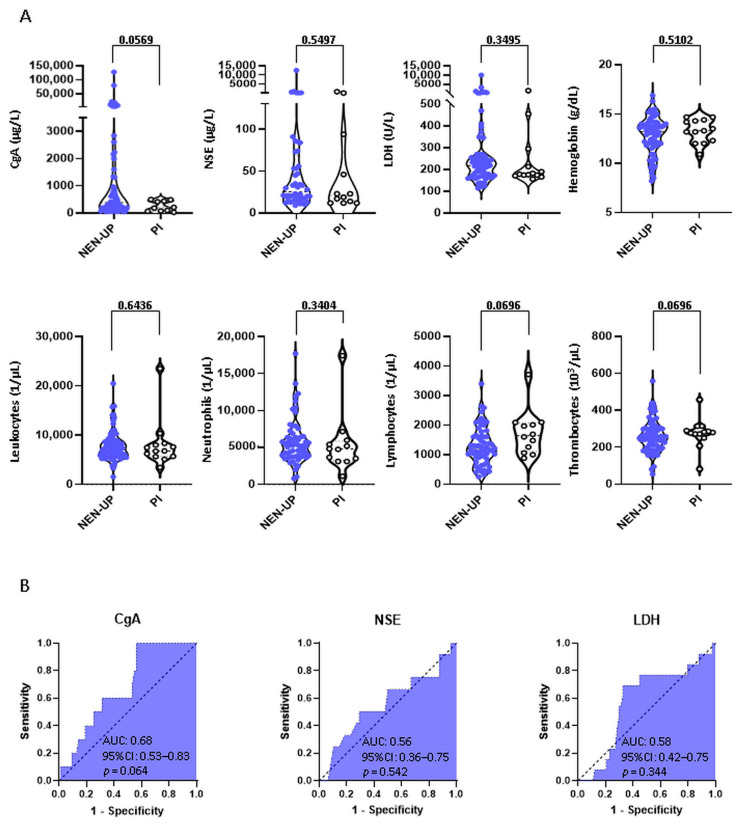
Contrast of laboratory parameters between patients with or without an identified primary tumor site. (**A**) Comparison of serum levels of CgA, NSE and LDH and WBC, ANC, ALC, Hb and PLT levels between patients of the NEN-UP and PI cohorts. (**B**) Predictive values of CgA, NSE and LDH for the detection of a primary tumor site were determined using ROC analysis. Abbreviations: NEN-UP = neuroendocrine neoplasm of unknown primary; PI = primary tumor site identified later on; CgA = chromogranin A; NSE = neuron-specific enolase; LDH = lactate dehydrogenase; AUC = area under the curve; CI = confidence interval.

**Figure 3 cancers-15-04316-f003:**
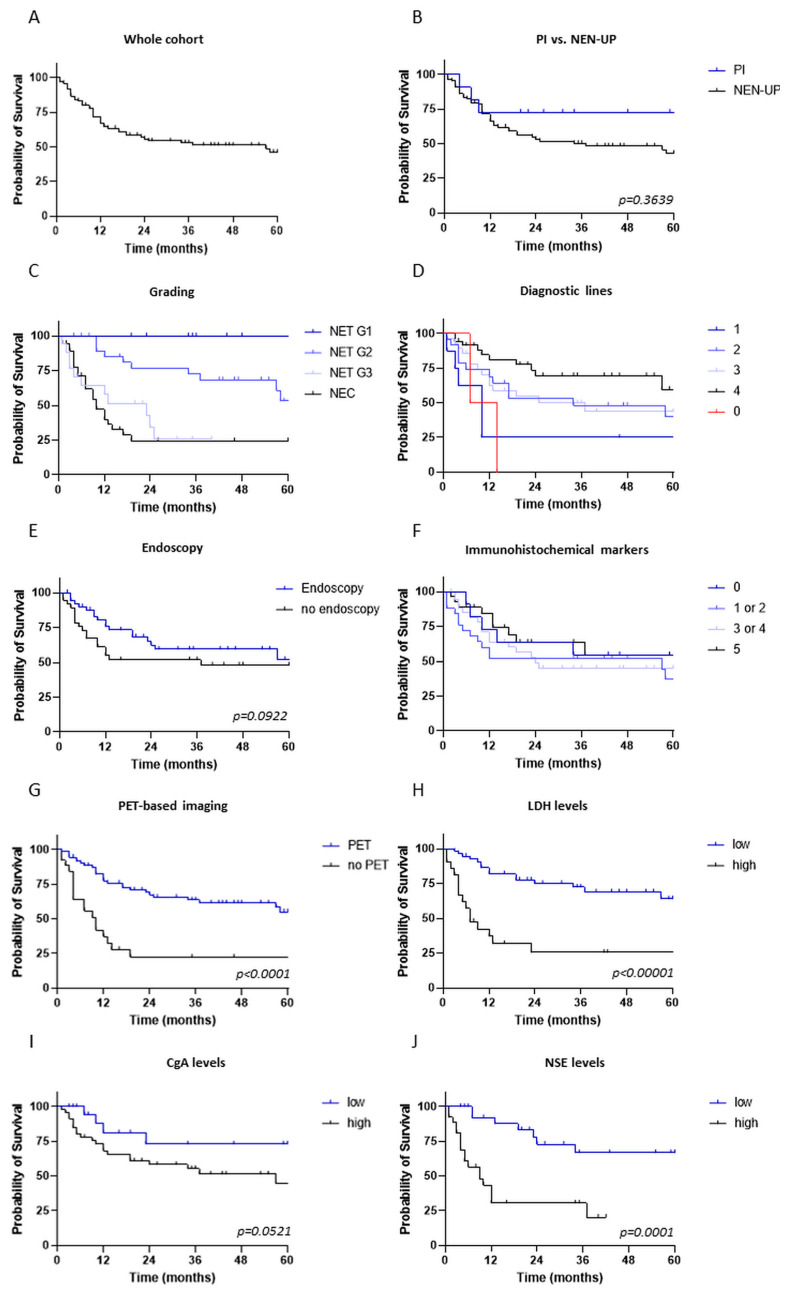
Prognostic role of clinical features, diagnostic instruments and therapeutic characteristics in patients with NEN-UP. Kaplan–Meier curves illustrate the probability of survival of the whole patient cohort (**A**), the PI and NEN-UP cohort (**B**) and depending on histological grading (**C**), the number of diagnostic lines (including serum tumor markers, immunohistochemistry, endoscopy and PET-based imaging) (**D**), the performance of endoscopy (**E**), immunohistochemistry (**F**) and PET-based imaging and (**G**) serum levels of LDH (**H**), CgA (**I**) and NSE (**J**). Abbreviations: NEN-UP = neuroendocrine neoplasm of unknown primary; PI = primary tumor site identified later on; CgA = chromogranin A; NSE = neuron-specific enolase; LDH = lactate dehydrogenase.

**Figure 4 cancers-15-04316-f004:**
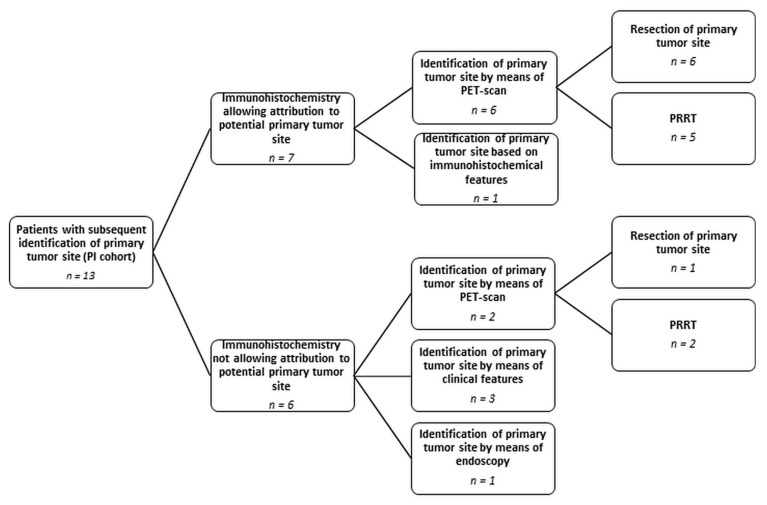
Analysis of patients with the subsequent identification of a primary tumor site (PI cohort). Modality of identifying the primary tumor site depending on immunohistochemical studies and resulting therapeutic consequences. Abbreviations: PI = primary tumor site identified later on; PRRT = peptide receptor radiotherapy.

**Table 1 cancers-15-04316-t001:** Patient characteristics.

	All Patients (%)	NEN-UP (%)	PI (%)	*p*-Value
number	113	100 (88.5)	13 (11.5)	
Sex				0.075
female	50 (44.2)	41 (41)	9 (69.2)	
male	63 (55.8)	59 (59)	4 (30.8)	
Age (in years)				0.2969
average	63.4	63.4	63.4	
<30	2 (1.8)	2 (2)	0 (0)	
30–39	5 (4.4)	3 (3)	2 (15.4)	
40–49	8 (7.1)	7 (7)	1 (7.7)	
50–59	23 (20.4)	20 (20)	3 (23.1)	
60–69	32 (28.3)	28 (28)	4 (30.8)	
70–79	35 (31.0)	33 (33)	2 (15.4)	
≥80	8 (7.1)	7 (7)	1 (7.7)	
Body-mass-index (kg/m^2^)				0.2725
average	25	25.3	23.4	
Histological Subtype				>0.9999
NET G1	16 (14.2)	15 (15)	1 (7.7)	
NET G2	36 (31.9)	32 (32)	4 (30.8)	
NET G3	18 (15.9)	15 (15)	3 (23.1)	
NEC	43 (38.1)	38 (38)	5 (38.5)	
Proliferation index				0.4614
Average (range)	36% (1–100%)	35% (1–100%)	36% (1–90%)	
Metastatic sites				0.7292
liver	77 (68.1)	65 (65)	12 (92.3)	
lung	26 (23)	24 (24)	2 (15.4)	
bones	33 (29.2)	30 (30)	3 (23.1)	
lymph nodes	61 (54)	54 (54)	7 (53.8)	
brain	8 (7.1)	8 (8)	0 (0)	
others	39 (34.5)	34 (34)	5 (38.5)	
Number of metastatic sites				
1	38 (33.6)	36 (36)	2 (15.4)	0.4225
2	34 (30.1)	29 (29)	5 (38.5)	
3	20 (17.7)	17 (17)	3 (23.1)	
4	15 (13.3)	12 (12)	3 (23.1)	
≥5	6 (5.3)	6 (6)	0 (0)	
Origin of initial histology				**0.0466**
liver	65 (57.5)	57 (57)	8 (61.5)	
lung	1 (0.8)	0 (0)	1 (7.7)	
genitourinary	6 (5.3)	4 (4)	2 (15.4)	
gastrointestinal	9 (8)	8 (8)	1 (7.7)	
skin	3 (2.7)	2 (2)	1 (7.7)	
lymph node	18 (15.9)	18 (18)	0 (0)	
central nervous system	4 (3.5)	4 (4)	0 (0)	
musculoskeletal	4 (3.5)	4 (4)	0 (0)	
others	3 (2.7)	3 (3)	0 (0)	
Date of initial diagnosis				0.1898
2010–2014	17 (15)	15 (15)	2 (15.4)	
2015–2019	51 (45.1)	48 (48)	3 (23.1)	
2020–2023	45 (39.8)	37 (37)	8 (61.5)	

Abbreviations: NEN-UP = neuroendocrine neoplasm of unknown primary; PI = primary tumor site identified later on; NET = neuroendocrine tumor; NEC = neuroendocrine carcinoma; *p*-value depicted in bold numbers: indicating statistical significance.

**Table 2 cancers-15-04316-t002:** Diagnostic work-up in patients with NEN of a known and unknown primary tumor site.

	All Patients (%)	NEN-UP (%)	PI (%)	*p*-Value
Number	113	100 (88.5)	13 (11.5)	
Bloodwork				
CgA	75 (66.4)	65 (65)	10 (76.9)	0.538
NSE	60 (53.1)	48 (48)	12 (92.3)	**0.0025**
CEA	28 (24.8)	26 (26)	2 (15.4)	0.5134
CA19-9	29 (25.7)	26 (26)	3 (23.1)	>0.9999
LDH	87 (77)	74 (74)	13 (100)	**0.037**
WBC	89 (78.8)	76 (76)	13 (100)	0.0669
ANC	73 (64.6)	61 (61)	12 (92.3)	**0.0307**
ALC	73 (64.6)	61 (61)	12 (92.3)	**0.0307**
Hb	89 (78.8)	76 (76)	13 (100)	0.0669
PLT	88 (77.9)	75 (75)	13 (100)	0.0686
Urinalysis				
24 h 5-hydroxyindoleacetic acid	6 (5.3)	6 (6)	0 (0)	>0.9999
Immunohistochemistry				
Synaptophysin	92 (81.4)	79 (79)	13 (100)	0.1223
CgA	74 (65.5)	62 (62)	12 (92.3)	**0.0326**
CD56	41 (36.3)	37 (37)	4 (30.8)	0.7664
CDX2	66 (58.4)	56 (56)	10 (76.9)	0.2319
TTF-1	76 (67.3)	68 (68)	8 (61.5)	0.7549
ISLET-1	40 (35.4)	34 (34)	6 (46.2)	0.5385
Serotonin	18 (15.9)	16 (16)	2 (15.4)	>0.9999
Gastrin	3 (2.7)	3 (3)	0 (0)	>0.9999
CK20	29 (25.7)	29 (29)	0 (0)	**0.0203**
Adherence to IHC guidelines				0.4942
Full adherence (5 out of 5 markers)	32 (28.3)	27 (27)	5 (38.5)	
Partial adherence (3 or 4 markers)	40 (35.4)	35 (35)	5 (38.5)	
Minimal adherence (1 or 2 markers)	27 (23.9)	24 (24)	3 (23.1)	
No adherence (no marker)	14 (12.4)	14 (14)	0 (0)	
Imaging				
FDG-PET	24 (21.2)	23 (23)	1 (7.7)	0.2935
SSTR-PET	71 (62.8)	62 (62)	9 (69.2)	0.7643
Endoscopy				0.6438
Gastroscopy	11 (9.7)	9 (9)	2 (15.4)	
Colonoscopy	7 (6.2)	6 (6)	1 (7.7)	
Gastroscopy + Colonoscopy	42 (37.2)	36 (36)	6 (46.2)	
no endoscopy	53 (46.9)	49 (49)	4 (30.8)	
NGS				
	31 (27.4)	26 (26)	5 (38.5)	0.3398
CIN				
	30 (26.5)	25 (25)	5 (38.5)	0.3253
Diagnostic lines ^§^				0.5673
0	3 (2.7)	3 (3)	0 (0)	
1	11 (9.7)	11 (11)	0 (0)	
2	25 (22.1)	23 (23)	2 (15.4)	
3	34 (30.1)	29 (29)	5 (38.5)	
4	40 (35.4)	34 (34)	6 (46.2)	

Abbreviations: NEN-UP = neuroendocrine neoplasm of unknown primary; PI = primary tumor site identified later on; CgA = chromogranin A; NSE = neuron-specific enolase; CEA = carcinoembryonic antigen; LDH = lactate dehydrogenase; WBC = white blood count; ANC = absolute neutrophil count; ALC = absolute lymphocyte count; Hb = hemoglobin; PLT = platelet; CK = cytokeratin; IHC = immunohistochemistry; FDG = fluorodeoxyglucose; CT = computed tomography; SSTR = somatostatin receptor; NGS = next-generation sequencing; CIN = chromosomal instability. ^§^: including serum tumor markers, immunohistochemistry, endoscopy and PET-based imaging. *p*-value depicted in bold numbers: indicating statistical significance.

**Table 3 cancers-15-04316-t003:** Diagnostic work-up in patients with NET and NEC of a known and unknown primary tumor site.

	All Patients (%)	NET	NEC	*p*-Value
Number	113	70 (61.9)	43 (38.1)	
Bloodwork				
CgA	75 (66.4)	53 (75.7)	22 (51.2)	**0.0132**
NSE	60 (53.1)	35 (50)	25 (58.1)	0.4416
CEA	28 (24.8)	15 (21.4)	13 (30.2)	0.3701
CA19-9	29 (25.7)	17 (24.3)	12 (27.9)	0.6649
LDH	87 (77)	55 (78.6)	32 (74.4)	0.6496
WBC	89 (78.8)	56 (80)	33 (76.7)	0.8132
ANC	73 (64.6)	42 (60)	31 (72.1)	0.2273
ALC	73 (64.6)	42 (60)	31 (72.1)	0.2273
Hb	89 (78.8)	56 (80)	33 (76.7)	0.8132
PLT	88 (77.9)	55 (78.6)	33 (76.7)	0.8197
Urinalysis				
24 h urinary 5-hydroxyindoleacetic acid	6 (5.3)	6 (8.6)	0 (0)	>0.9999
Immunohistochemistry				
Synaptophysin	92 (81.4)	55 (78.6)	37 (86)	0.4557
CgA	74 (65.5)	49 (62)	25 (58.1)	0.2251
CD56	41 (36.3)	18 (25.7)	23 (53.5)	**0.0052**
CDX2	66 (58.4)	44 (62.9)	22 (51.2)	0.2432
TTF-1	76 (67.3)	45 (64.3)	31 (72.1)	0.4172
ISLET-1	40 (35.4)	30 (42.9)	10 (23.3)	**0.0432**
Serotonin	18 (15.9)	16 (22.9)	2 (4.7)	**0.0152**
Gastrin	3 (2.7)	3 (4.9)	0 (0)	0.2865
CK20	29 (25.7)	14 (20)	15 (34.9)	0.1195
Adherence to IHC guidelines				0.0797
Full adherence (5 out of 5 markers)	32 (28.3)	25 (35.7)	7 (16.3)	
Partial adherence (3 or 4 markers)	40 (35.4)	21 (30)	19 (44.2)	
Minimal adherence (1 or 2 markers)	27 (23.9)	14 (20)	13 (30.2)	
No adherence (no marker)	14 (12.4)	10 (14.3)	4 (9.3)	
Imaging				
FDG-PET	24 (21.2)	8 (11.4)	16 (37.2)	**0.0011**
SSTR-PET	71 (62.8)	60 (85.7)	11 (25.6)	**<0.0001**
Endoscopy				0.6482
Gastroscopy	11 (9.7)	8 (11.4)	3 (7.0)	
Colonoscopy	7 (6.2)	4 (5.7)	3 (7.0)	
Gastroscopy + Colonoscopy	42 (37.2)	28 (40)	14 (32.6)	
no endoscopy	53 (46.9)	30 (42.9)	23 (53.5)	
NGS				
	31 (27.4)	15 (21.4)	16 (37.2)	0.0679
CIN				
	30 (26.5)	15 (21.4)	15 (34.9)	0.1296
Diagnostic lines ^§^				**0.0334**
0	3 (2.7)	0 (0)	3 (7)	
1	11 (9.7)	6 (8.6)	5 (11.6)	
2	25 (22.1)	15 (21.4)	10 (23.3)	
3	34 (30.1)	18 (25.7)	16 (37.2)	
4	40 (35.4)	31 (44.3)	9 (20.9)	

Abbreviations: NET = neuroendocrine tumor; NEC = neuroendocrine carcinoma; CgA = chromogranin A; NSE = neuron-specific enolase; CEA = carcinoembryonic antigen; LDH = lactate dehydrogenase; WBC = white blood count; ANC = absolute neutrophil count; ALC = absolute lymphocyte count; Hb = hemoglobin; PLT = platelet; CK = cytokeratin; IHC = immunohistochemistry; FDG = fluorodeoxyglucose; CT = computed tomography; SSTR = somatostatin receptor; NGS = next-generation sequencing; CIN = chromosomal instability. ^§^: including serum tumor markers, immunohistochemistry, endoscopy and PET-based imaging. *p*-value depicted in bold numbers: indicating statistical significance.

**Table 4 cancers-15-04316-t004:** Univariable logistic regression analysis between different diagnostic approaches and the identification of a primary tumor site.

Diagnostic Test	Odds Ratio (95% CI)	*p*-Value
Blood work		
CgA	1.80 (0.51–8.39)	0.397
NSE	13 (2.42–241.2)	**0.016**
LDH	x	x
Endoscopy		
Any form of endoscopy	2.16 (0.66–8.40)	0.224
Gastroscopy	1.84 (0.26–8.37)	0.471
Colonoscopy	1.31 (0.07–8.60)	0.812
Gastroscopy + Colonoscopy	1.52 (0.46–4.93)	0.478
Imaging		
PET-based imaging	0.83 (0.25–3.27)	0.775
SSTR-PET	1.38 (0.42–5.37)	0.613
FDG-PET	x	x
SSTR- and FDG-PET	0.61 (0.03–3.56)	0.65
Diagnostic Lines	1.67 (0.92–3.47)	0.122
Immunohistochemistry		
Adherence to IHC guidelines	1.91 (1.21–3.48)	**0.015**
Synaptophysin	x	x
CgA	7.36 (1.37–136.6)	0.06
CD56	0.757 (0.19–2.50)	0.661
CDX2	2.62 (0.75–12.2)	0.162
TTF-1	0.753 (0.23–2.66)	0.641
ISLET-1	1.664 (0.50–5.39)	0.392
Serotonin	0.955 (0.14–4.01)	0.955
Gastrin	x	x
CK20	x	x
NGS	1.78 (0.50–5.83)	0.348

Abbreviations: CgA = chromogranin A; NSE = neuron-specific enolase; LDH = lactate dehydrogenase; IHC = immunohistochemistry; CK = cytokeratin; NGS = next-generation sequencing; x: not applicable. *p*-value depicted in bold numbers: indicating statistical significance.

**Table 5 cancers-15-04316-t005:** Therapeutic approaches.

	All Patients (%)	NEN-UP (%)	PI (%)	*p*-Value
Treatment regimen				
Systemic Therapy				
Chemotherapy	58 (51.3)	49 (49)	9 (69.2)	0.24
SSA	42 (37.2)	35 (35)	7 (53.8)	0.2274
Radiation Therapy				
Local radiotherapy	26 (23.0)	24 (24)	2 (15.4)	0.7291
PRRT	25 (22.1)	18 (18)	7 (53.8)	**0.008**
SIRT	4 (3.5)	3 (3)	1 (7.7)	0.391
Local therapy				
Surgery	39 (34.5)	32 (32)	7 (53.8)	0.1326
RFA	1 (0.9)	1 (1)	0 (0)	>0.9999
BSC	8 (7.1)	7 (7)	1 (7.7)	>0.9999
Lines of systemic therapy				**0.0127**
1	55 (48.7)	51 (51)	4 (30.8)	
2	17 (15)	11 (11)	6 (46.2)	
3	14 (12.4)	13 (13)	1 (7.7)	
4+	8 (4.4)	5 (5)	0 (0)	

Abbreviations: NEN-UP = neuroendocrine neoplasm of unknown primary; PI = primary tumor site identified later on; SSA = somatostatin analogue; PRRT = peptide receptor radiotherapy; SIRT = selective internal radiation therapy; RFA = radiofrequency ablation; BSC = best supportive care; FU = fluorouracil. *p*-value depicted in bold numbers: indicating statistical significance.

## Data Availability

The data presented in this study are available on request from the corresponding author.

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
