# Peer review of "Diagnostic Approaches for Neuroendocrine Neoplasms of Unknown Primary (NEN-UPs) and Their Prognostic Relevance—A Retrospective, Long-Term Single-Center Experience"

_cancers, 2023, doi:10.3390/cancers15174316_

Round 1
Reviewer 1 Report
This is a well-written paper trying to assess NEN-UPs from clinical, histological and immunohistochemical angles. The clinical need for studies like this is significant, since a large subset of patients may present with a metastasis and no known primary site - causing difficulites in establishing the most appropriate treatment for each patient. The written text flows well, and the illustrations are nice additions, although they need improvements. I will list these issues, plus additional queires that need attention:
1. There is no information as to how the Ki-67 labeling index was retrieved. From manual counting of 2,000 tumor cells using an ocular grid and hotspot areas? Given the rapid implementation of digital pathology, this information is crucial. Antibody clone?
2. Also, please state if the mitotic index was assessed in this cohort, and if so - whether or not this parameter correlated to the Ki-67 index or not. Since this is a retrospective cohort ranging from 2010-23, this would ensure retained immunoreactivity in the FFPE blocks that were >10 years old. Or was antigenicity proven by other means?
3. The authors make a point of ISLET1 (ISL1) as being a marker of pancreatic NEN, but care must be taken here - as this marker has been found positive also in non-pancreatic NENs. A seminal paper is found at PMID: 32813226 and should be referenced, and the drawbacks of using ISLET1 as a pancreatic marker instead of for example PDX1 must be discussed.
4. CK20 is not a perfect identifier for Merkel cell carcinoma (row 218), so this sentence should be revised.
5. 12% of patients did not undergo IHC for the neuroendocrine markers CgA, SYP, ISL1, CD56 (rows 222-223)? Then how were they determined as NENs in the first place? And if the study is performed in retrospect, how come they were not stained for the purpose of this study?
6. The inclusion of NGS data is beyond me, as no results from this analysis are presented, only information whether or not it was performed? Would it be fair to state that the inclusion of important mutations should be a pert of this paper, or is it redundant?
Minor issues:
Table 1: Gender, change to "sex", as some believe the former terminology is subjective.
Table 1: Why use "CUP" as the preferred terminology, when so many of these cases were not carcinomas, but rather low-grade NENs?
Table 1: What does the numbers 1-5 stand for in the leftmost column? Number of metastatic sites? Why not state so?
Figure 1: CUP-NEN? What is that? Please be consistent with the terminology (NEN-UPs, CUPs et.c.).
Figure 1: What is "lymph node invasion"? Do you mean "lymph node metastasis"?
Figure 1D needs revisions: What is "PI" (?) in white? Primary identified (as in the main text?). Please write out all abbreviations in full in all figure legends. Also, some NEN-UPs have no metastases? How are they then identified as NEN-UPs in the first place? Could this be clarified?
Figure 4 could be better explained (it only involves PI cases).
Author Response
Reviewer 1
This is a well-written paper trying to assess NEN-UPs from clinical, histological and immunohistochemical angles. The clinical need for studies like this is significant, since a large subset of patients may present with a metastasis and no known primary site - causing difficulties in establishing the most appropriate treatment for each patient. The written text flows well, and the illustrations are nice additions, although they need improvements. I will list these issues, plus additional queries that need attention:
- There is no information as to how the Ki-67 labeling index was retrieved. From manual counting of 2,000 tumor cells using an ocular grid and hotspot areas? Given the rapid implementation of digital pathology, this information is crucial. Antibody clone?
We thank the reviewer for this comment. For those cases being diagnosed at the Pathology Institute of the University Hospital Tuebingen (~50%) manual counting of hotspot areas was performed and since 2021 also digital Ki67 evaluation (using Cognition Master, VMscope GmbH, Germany). We used the Ki67 (clone Mib1) antibody from Agilent Technologies (#M724, 1:400).
To reflect this issue, the manuscript was amended accordingly:
Page 3, lines 136-140.
- Also, please state if the mitotic index was assessed in this cohort, and if so - whether or not this parameter correlated to the Ki-67 index or not. Since this is a retrospective cohort ranging from 2010-23, this would ensure retained immunoreactivity in the FFPE blocks that were >10 years old. Or was antigenicity proven by other means?
We thank the reviewer for raising this issue. The mitotic index was not routinely assessed (except for pulmonary typical and atypical carcinoids). However, the Ki67 values used in the study were taken from the pathology reports and thus were determined at the time of diagnosis on sections from “freshly” prepared FFPE blocks with retained immunoreactivity.
To reflect this issue, the manuscript was amended accordingly:
Page 3, lines 140-141.
- The authors make a point of ISLET1 (ISL1) as being a marker of pancreatic NEN, but care must be taken here - as this marker has been found positive also in non-pancreatic NENs. A seminal paper is found at PMID: 32813226 and should be referenced, and the drawbacks of using ISLET1 as a pancreatic marker instead of for example PDX1 must be discussed.
We thank the reviewer for pointing us towards this important issue. It is absolutely true that ISLET-1 is not a pancreas-specific immunohistochemical marker – Agaimy et al. reported the expression of ISLET-1 in poorly differentiated extrapancreatic NENs, like for example in neuroendocrine carcinomas of the lungs, the head and neck, or in Merkel cell carcinomas. Furthermore, poorly differentiated pancreatic NENs show a reduced expression of ISLET-1 compared to well differentiated NENs of pancreatic origin (Agaimy et al., Modern Pathology 2013). More recently, Juhlin et al. emphasized the role of ISLET-1 as a general marker of neuroendocrine differentiation, especially in high-grade NECs (Juhlin et al., Endocrine Pathology 2020). Immunohistochemical markers such as PDX1, PAX8 or NESP55 are considered to be more specific for NENs of pancreatic origin (Bocchini et al., Frontiers in Oncology 2020).
In our study cohort about 50% of the patient cases have been histopathologically diagnosed at the Pathology Institute of the University Hospital Tuebingen. The other 50% of cases were initially diagnosed by external referring physicians. However, no other markers besides ISLET-1 have been universally established and available by default that have a higher specificity for pancreatic NENs. Due to the retrospective character of this study, no additional markers were subsequently analyzed.
To reflect these results, the manuscript was amended accordingly:
Page 21, lines 513-526.
- CK20 is not a perfect identifier for Merkel cell carcinoma (row 218), so this sentence should be revised.
We thank the reviewer for this insightful advice and fully agree with his or her statement.
CK20 is expressed by a large number of Merkel cell carcinomas; however, cases of Merkel cell carcinoma without CK20 expression have been reported (Miner et al., Modern Pathology 2015). In addition, CK20 has limited specificity as an immunohistochemical marker, since it can also be found in colorectal, gastric, or pancreatic carcinomas, among others (Bayrak et al., Diagnostic Pathology 2012). Here, CK20 was seen in a screening panel only as a possible hint for the presence of Merkel cell carcinoma.
To reflect these results, the manuscript was amended accordingly:
Page 8, lines 238-239.
- 12% of patients did not undergo IHC for the neuroendocrine markers CgA, SYP, ISL1, CD56 (rows 222-223)? Then how were they determined as NENs in the first place? And if the study is performed in retrospect, how come they were not stained for the purpose of this study?
We thank the reviewer for raising this issue. We acknowledge that in 12% of all patients with NEN-UPs no immunohistochemistry at initial diagnosis was documented in our system. 85% of these patients did not receive their histological diagnosis at the University Hospital Tuebingen and subsequently presented for further treatment planning at our center. Even despite of our great effort to obtain all histopathological data, no dedicated IHC staining including CgA, SYP, ISL1, CD56, even if performed elsewhere, could be verified. Only one of these patients was initially diagnosed at the University Hospital Tuebingen. In this case unfortunately, the biopsy was not suitable for extensive IHC analysis. As a result, the initial diagnosis was based on morphological assessment in histopathology in combination with clinical, laboratory, and imaging findings.
In patients who didn’t receive a dedicated IHC analysis to further identify the primary organ three possible reasons were identified: I) Not enough tumor material. II) Lack of clinical evidence for a NEN-UP. III) Conservation of the material for molecular pathology.
The primary objective of this retrospective study was to assess the extent of diagnostic work-up in this cohort of patients initially diagnosed with NEN-UP. In the next step, conclusions on the clinical outcome and a potential identification of the primary tumor site should be drawn from this. Following the retrospective character of this study, we aimed to portray a status regarding the diagnostic work-up performed in this patient cohort in the period from 2010 to 2023. Our primary focus was not on improving these conditions.
However, this is an important finding of our study, which needs to be considered and optimized in the future. This seems particularly important as we could show that a detailed immunohistochemical examination was associated with the identification of a primary tumor site which is beneficial for the clinical outcome of patients with NEN-UP. This cohort of patients who did not receive immunohistochemical analysis at initial diagnosis show a median overall survival (OS) of 43.5 months and a primary tumor site could not be identified in any of these patients.
To reflect these thoughts, the manuscript was amended accordingly:
Page 8, lines 244-246.
Page 20/21, lines 494-506.
- The inclusion of NGS data is beyond me, as no results from this analysis are presented, only information whether or not it was performed? Would it be fair to state that the inclusion of important mutations should be a part of this paper, or is it redundant?
We thank the reviewer for raising this issue. Integration of NGS into standard diagnostic algorithm can help improve the diagnosis of NEN-UPs (Venizelos et al., Endocrine-Related Cancer 2021). On the one hand, the identification of specific genetic signatures by means of NGS may enable the assignment of a NEN to a certain tissue of origin, especially in NECs. On the other hand, genetic alterations that are of therapeutic relevance can be detected and can form the basis of targeted treatment approaches.
At the University Hospital Tuebingen, NGS has been increasingly performed in recent years. 74% of patients who underwent NGS were diagnosed in the years 2019 or later. This corresponds to approximately 38% of all patients with NEN-UP who presented at our center during this period which underlines the need of NGS in the future.
The results of the performed NGS analyses in this patient cohort are depicted in Supplementary Table S2 and the manuscript was modified accordingly:
Page 8, line 253-254.
Minor issues:
Table 1: Gender, change to "sex", as some believe the former terminology is subjective.
We thank the reviewer for pointing us towards this important issue.
Table 1 was modified accordingly.
Table 1: Why use "CUP" as the preferred terminology, when so many of these cases were not carcinomas, but rather low-grade NENs?
We thank the reviewer for this rightful and important comment.
The terminology was changed to “NEN-UP”:
Figure 1, 2 and 3 were modified accordingly.
Table 1, 2, 5 and Supplementary Table S1 were modified accordingly.
The manuscript was modified accordingly:
Page 3, line 149.
Page 4, line 163, 164, 169, 177, 199/200.
Page 5, line 204.
Page 8, line 231, 263 and 265.
Page 15, line 353.
Page 16, line 371, 374 and 413.
Page 23, line 627.
Table 1: What does the numbers 1-5 stand for in the leftmost column? Number of metastatic sites? Why not state so?
We thank the reviewer for this insightful advice and fully agree with his or her statement.
These numbers indicate the number of metastatic sites.
Table 1 was modified accordingly.
Figure 1: CUP-NEN? What is that? Please be consistent with the terminology (NEN-UPs, CUPs etc.).
We thank the reviewer for pointing us towards this important issue.
Figure 1 was modified accordingly.
Figure 1: What is "lymph node invasion"? Do you mean "lymph node metastasis"?
We thank the reviewer for this rightful and important comment.
In this context, we referred to lymph node metastasis and Figure 1 was modified accordingly.
Figure 1D needs revisions: What is "PI" (?) in white? Primary identified (as in the main text?). Please write out all abbreviations in full in all figure legends. Also, some NEN-UPs have no metastases? How are they then identified as NEN-UPs in the first place? Could this be clarified?
We thank the reviewer for raising this issue and apologize for the lack of clear explanation of the abbreviations used in the figures.
Here, PI has the same meaning as in the main text and means “Primary identified”.
An explanation of the abbreviations used in Figure 1, 2, 3 and 4 as well as in Supplementary Figure S1 was added to the figure legends accordingly.
Furthermore, the group of NEN-UP patients lacking distant metastases include cases with evidence of a single NEN at a body site which could not be identified as the origin of the NEN by means of clinical or histological investigations. This comprises all patients showing only lymph node metastases or singular manifestations, for example in the soft tissue.
To reflect these thoughts, the manuscript was amended accordingly:
Page 3, line 122-125.
Figure 4 could be better explained (it only involves PI cases).
We thank the reviewer for this rightful and important comment.
Figure 4 and its caption were modified accordingly.
Reviewer 2 Report
In this article, the authors analyzed the diagnostic work-up performed in 113 patients with NEN-UPs and characterized the relationship between certain diagnostic work-up and identification of the primary tumor site. The manuscript is straightforward, well written, and concise and has clear results within the scope of a retrospective analysis. Definitely deserves to be published and is a valuable contribution to the “cancers” journal. Some minor comments need to be addressed before publication, as follows.
[1] “1. Introduction”, Page 2 of 26, Lines 67-70:
“On top of that, CUP is one of the leading malignancies responsible for cancer-related mortality and the majority of patients diagnosed with CUP show an unfavorable prognosis with a median overall survival (OS) of only six to 12 months [11-19].”.
The authors should clarify that patients with CUP are categorized into two prognostic subgroups, according to their clinicopathologic characteristics. The majority of patients with CUP (80–85%) belong to unfavorable subsets. The favorable risk cancer subgroup (15–20 %) includes patients with neuroendocrine carcinomas of unknown primary, peritoneal adenocarcinomatosis of a serous papillary subtype, isolated axillary nodal metastases in females, squamous cell carcinoma involving non-supraclavicular cervical lymph nodes, single metastatic deposit from unknown primary and men with blastic bone metastases and PSA expression. Very recently, new favorable subsets of CUP seem to emerge including colorectal, lung and renal CUP which underlies specific treatments.
Recommended reference: Rassy E, et al. New rising entities in cancer of unknown primary: Is there a real therapeutic benefit? Crit Rev Oncol Hematol. 2020 Mar;147:102882.
[2] “Table 2”, Page 8 of 26:
From the therapeutic point of view, chromosomal instability (CIN) is not a frequent phenomenon in CUP, which may favour immune checkpoint inhibitors (ICI) among patients with CUP. Conversely, these patients present individual gene alterations implicated in immune-evasion and resistance to ICI. Have the authors considered CIN in their study? That information – if available – could be incorporated in table 2.
[3] “4. Discussion”, Page 17 of 26, Lines 413-416:
“The detection rate of a primary tumor site in patients originally being classified as a CUP is known to vary between 10 and 40% according to the literature and could be increased more recently by means of newly available diagnostic tools, including NGS or molecular profiling [30-34].”.
Furthermore, the authors should mention that in the era of targeted therapies, accurate histopathological and molecular classification of tumours is essential, in order to administer the best tailored therapeutic strategy. Classifications based on epigenetic alterations have served this purpose. Indeed, cancer cells are characterized by a massive overall loss of DNA methylation (20–60% overall decrease in 5-methylcytosine), and by the simultaneous acquisition of specific patterns of hypermethylation at CpG islands of certain promoters, which can reversibly or irreversibly alter gene function, thereby contributing to cancer progression.
Recommended reference: Moran S, et al. Precision medicine based on epigenomics: the paradigm of carcinoma of unknown primary. Nat Rev Clin Oncol. 2017 Nov;14(11):682-694.
[4] “4. Discussion”, Page 19 of 26, Lines 524-527:
“In NENs, histologic origin has a strong impact on the therapy strategy since certain treatment modalities are only available for distinct subentities leading to a superior clinical outcome in these patients [20].”.
At that stage, the authors are highly encouraged to report that there are significant deficiencies in the currently available studies comparing site-specific therapy and empiric chemotherapy in CUP patients. The assessment of the literature allows to recommend two comprehensive clinical trial designs, a visionary and a pragmatic approach. Both are amenable to implementing the latest diagnostics and therapeutic advances to improve the quality of CUP research and the prognosis of many patients.
Recommended reference: Rassy E, et al. Systematic review of the CUP trials characteristics and perspectives for next-generation studies. Cancer Treat Rev. 2022 Jun;107:102407.
Author Response
Reviewer 2
In this article, the authors analyzed the diagnostic work-up performed in 113 patients with NEN-UPs and characterized the relationship between certain diagnostic work-up and identification of the primary tumor site. The manuscript is straightforward, well written, and concise and has clear results within the scope of a retrospective analysis. Definitely deserves to be published and is a valuable contribution to the “cancers” journal. Some minor comments need to be addressed before publication, as follows.
[1] “1. Introduction”, Page 2 of 26, Lines 67-70:
“On top of that, CUP is one of the leading malignancies responsible for cancer-related mortality and the majority of patients diagnosed with CUP show an unfavorable prognosis with a median overall survival (OS) of only six to 12 months [11-19].”.
The authors should clarify that patients with CUP are categorized into two prognostic subgroups, according to their clinicopathologic characteristics. The majority of patients with CUP (80–85%) belong to unfavorable subsets. The favorable risk cancer subgroup (15–20 %) includes patients with neuroendocrine carcinomas of unknown primary, peritoneal adenocarcinomatosis of a serous papillary subtype, isolated axillary nodal metastases in females, squamous cell carcinoma involving non-supraclavicular cervical lymph nodes, single metastatic deposit from unknown primary and men with blastic bone metastases and PSA expression. Very recently, new favorable subsets of CUP seem to emerge including colorectal, lung and renal CUP which underlies specific treatments.
Recommended reference: Rassy E, et al. New rising entities in cancer of unknown primary: Is there a real therapeutic benefit? Crit Rev Oncol Hematol. 2020 Mar;147:102882.
We thank the reviewer for pointing us towards this important issue. The assignment of CUP patients either to the group with favorable or unfavorable clinical course is of therapeutic as well as of prognostic importance. This concept and its latest developments are now addressed in the introduction.
The manuscript was modified accordingly:
Page 2, line 64-79.
[2] “Table 2”, Page 8 of 26:
From the therapeutic point of view, chromosomal instability (CIN) is not a frequent phenomenon in CUP, which may favour immune checkpoint inhibitors (ICI) among patients with CUP. Conversely, these patients present individual gene alterations implicated in immune-evasion and resistance to ICI. Have the authors considered CIN in their study? That information – if available – could be incorporated in table 2.
We thank the reviewer for raising this issue.
In this patient cohort, analysis of chromosomal instability (CIN) was performed in 30 patients (26.5%). Among these patients, 17 patients showed CIN, comprising 12 patients diagnosed with neuroendocrine carcinoma (NEC) and 5 patients diagnosed with neuroendocrine tumor (NET) – 2 patients with NET G2 and 3 patients with NET G3. Comparing patients of the “NEN-UP (neuroendocrine neoplasm of unknown primary)” and “PI (primary identified)” cohort, 5 out of 13 patients (38%) belonging to the PI and 25 patients out of 100 patients (25%) belonging to the NEN-UP cohort were assessed with regard to CIN. Thus, in 2 patients of the PI (15%) and in 15 patients of the NEN-UP cohort (15%) CIN was detected.
No statistically significant difference with regard to the performance rate of analysis of CIN could be found between the cohorts NEN-UP and PI on the one hand or NET and NEC on the other hand.
This information was added to Table 2 and 3 accordingly.
The manuscript was modified accordingly:
Page 3, line 144.
Page 4, line 168.
Page 7, line 224.
Page 8, line 253-254.
[3] “4. Discussion”, Page 17 of 26, Lines 413-416:
“The detection rate of a primary tumor site in patients originally being classified as a CUP is known to vary between 10 and 40% according to the literature and could be increased more recently by means of newly available diagnostic tools, including NGS or molecular profiling [30-34].”.
Furthermore, the authors should mention that in the era of targeted therapies, accurate histopathological and molecular classification of tumours is essential, in order to administer the best tailored therapeutic strategy. Classifications based on epigenetic alterations have served this purpose. Indeed, cancer cells are characterized by a massive overall loss of DNA methylation (20–60% overall decrease in 5-methylcytosine), and by the simultaneous acquisition of specific patterns of hypermethylation at CpG islands of certain promoters, which can reversibly or irreversibly alter gene function, thereby contributing to cancer progression.
Recommended reference: Moran S, et al. Precision medicine based on epigenomics: the paradigm of carcinoma of unknown primary. Nat Rev Clin Oncol. 2017 Nov;14(11):682-694.
We thank the reviewer for pointing us towards this important issue.
The role of epigenetic profiling as novel diagnostic tool for patients with cancer of unknown primary (CUP) is now addressed in the discussion and the manuscript was modified accordingly:
Page 19/20, line 450-460.
[4] “4. Discussion”, Page 19 of 26, Lines 524-527:
“In NENs, histologic origin has a strong impact on the therapy strategy since certain treatment modalities are only available for distinct subentities leading to a superior clinical outcome in these patients [20].”.
At that stage, the authors are highly encouraged to report that there are significant deficiencies in the currently available studies comparing site-specific therapy and empiric chemotherapy in CUP patients. The assessment of the literature allows to recommend two comprehensive clinical trial designs, a visionary and a pragmatic approach. Both are amenable to implementing the latest diagnostics and therapeutic advances to improve the quality of CUP research and the prognosis of many patients.
Recommended reference: Rassy E, et al. Systematic review of the CUP trials characteristics and perspectives for next-generation studies. Cancer Treat Rev. 2022 Jun;107:102407.
We thank the reviewer for this insightful advice and fully agree with his statement.
The challenges that arise in designing clinical trials to evaluate different therapeutic regimens in CUP patients and potential solutions are now addressed in the discussion and the manuscript was modified accordingly:
Page 22, line 598-605.
Reviewer 3 Report
Interesting study. The two main drawbacks are the retrospective design and the limited sample size. The authors should discuss on this.
Quality of the figures should be improved. They are too small and hard to read.
The authors should comment on the importance of the fact that primitive is unknown in these patients and this aspect should be adequately discussed in light of the well known benefit due to the resection of primary tumor resection in NENs (cite the recent series PMID: 27956320)
Author Response
Reviewer 3
Interesting study. The two main drawbacks are the retrospective design and the limited sample size. The authors should discuss on this.
We thank the reviewer for this rightful and important comment.
Limitations of the study, including the retrospective design as well as the limited number of patients analyzed, are now addressed as a separate section of the discussion.
The manuscript was amended accordingly:
Page 23/24, line 660-676.
Quality of the figures should be improved. They are too small and hard to read.
We thank the reviewer for this rightful and important comment.
Figure 1, 2, 3, 4 and Supplementary Figure S1 were modified accordingly.
The authors should comment on the importance of the fact that primitive is unknown in these patients and this aspect should be adequately discussed in light of the well known benefit due to the resection of primary tumor resection in NENs (cite the recent series PMID: 27956320).
We thank the reviewer for pointing us towards this important issue.
An unknown primary tumor complicates treatment and negatively affects the outcome of patients with NENs. Previous studies have investigated the extent to which surgical removal of the primary tumor in patients with mainly hepatic metastatic NENs of various origins affects patient survival (Capurso et al., Neuroendocrinology 2011; Bertani et al., Surgery 2014; Citterio et al., European Journal of Surgical Oncology 2017). These studies demonstrated that, despite the presence of distant metastases, removal of the primary tumor has a favorable effect on the survival of patients with NENs.
To reflect these results, the manuscript was amended accordingly:
Page 23, line 643-654.
Round 2
Reviewer 1 Report
Thank you for an excellent revision.
Reviewer 3 Report
The revised version of the paper is OK. Thank you!